# Impact of the COVID-19 Pandemic on Patients with Parkinson’s Disease from the Perspective of Treating Physicians—A Nationwide Cross-Sectional Study

**DOI:** 10.3390/brainsci12030353

**Published:** 2022-03-05

**Authors:** Andreas Wolfgang Wolff, Bernhard Haller, Antonia Franziska Demleitner, Erica Westenberg, Paul Lingor

**Affiliations:** 1Department of Neurology, Klinikum rechts der Isar, School of Medicine, Technical University of Munich, 81675 Munich, Germany; andreas.wolff@tum.de (A.W.W.); antonia.demleitner@tum.de (A.F.D.); erica.westenberg@tum.de (E.W.); 2Institute of AI and Informatics in Medicine, Klinikum rechts der Isar, School of Medicine, Technical University of Munich, 81675 Munich, Germany; bernhard.haller@tum.de; 3German Center for Neurodegenerative Diseases (DZNE), 81377 Munich, Germany

**Keywords:** Parkinson’s disease, COVID-19, provision of care, telemedicine

## Abstract

The COVID-19 pandemic has posed challenges to maintaining medical care for patients with Parkinson’s disease (PD). The Parkinson’s Disease during the COVID-19 Pandemic (ParCoPa) survey was conducted as an online, nationwide, cross-sectional survey from December 2020 to March 2021 and aimed to assess the impact of the pandemic on the medical care of PD patients from the physicians’ perspective. Invitations containing a randomly generated registration code were mailed to healthcare professionals from sixty-seven specialty centers in Germany. Confounders for the worsening of subjective treatment quality, perceived health risk due to the profession, and adequate protective measures against SARS-CoV-2 were assessed using logistic regression analysis. Of all forty physicians who responded, 87.5% reported a worsening of motor and nonmotor symptoms in their patients, 97.5% experienced cancellation of appointments, and difficulties in organizing advanced and supplementary therapies were reported by over 95%. Participants offered alternative consultation options, mostly in the form of telephone (77.5%) or online (64.1%) consultations, but telephone consultations were the most accepted by patients (“broadly accepted”, 40.0%). We identified pandemic-related deficits in providing care for patients with PD and areas of improvement to ensure continued care for this vulnerable patient population.

## 1. Introduction

The current corona virus disease 2019 (COVID-19) pandemic is posing a persistent challenge to our healthcare systems [1]. As of February 2022, the World Health Organization recorded over 386 million confirmed cases worldwide including around 10.6 million of them in Germany [2]. Age, male sex, and obesity have been identified as risk factors for a more severe disease course of COVID-19, resulting in an elevated risk for ICU treatment, respiratory failure, and death [3,4].

A recent nationwide analysis of hospitalized patients with Parkinson’s disease (PD) in Germany showed higher infection rates in this patient group compared to non-PD patients [5]. Moreover, infection of PD patients with SARS-CoV-2 is associated with a significant worsening of motor and nonmotor symptoms, and PD patients have an elevated risk for a fatal outcome compared to non-PD patients [5,6]. However, PD patients are not only affected by the risk of infection. As PD is a chronic progressive disease, patients require regular outpatient visits to a movement disorder specialist to adjust the treatment of their symptoms. In addition to pharmacological treatment, PD patients need frequent supportive therapies to conserve mobility, prevent falls, and train functions such as swallowing or speech [7].

Restrictions on activities of daily life during the pandemic, such as home confinement or travel restrictions, have posed risks to regular medical care for PD patients [8]. Difficulties obtaining medication due to quarantine measures [9], as well as a significant decrease in multimodal complex treatments and implementation of advanced pump therapies [10], may have contributed to an increased disease burden for PD patients. Reduction of exercise [7,11,12] and social isolation [13] also correlated with the worsening of PD symptoms. Accompanying these disruptions to medical care, various cross-sectional assessments of symptom burden have found significant symptom deterioration in these patients [8,14,15,16,17,18,19].

The current pandemic has severely impacted PD patients’ clinical routine. It is therefore imperative to identify deficits in patient care and develop possible strategies for improvement. As such, our nationwide online survey of specialists involved in the treatment of PD patients in Germany sought to characterize the situation of these patients and to identify challenges in the treatment of PD patients during the pandemic.

## 2. Materials and Methods

### 2.1. Study Design, Questionnaire, and Participants

The Parkinson’s disease during the COVID-19 Pandemic (ParCoPa) study was planned as a cross-sectional survey covering various aspects of PD patient care during the COVID-19 pandemic. Physicians from 67 specialist centers, 59 of which are part of the German Competence Network for Parkinson’s disease (https://www.kompetenznetz-parkinson.de/klinische-zentren, accessed 10 November 2020), were invited via sealed envelopes sent by mail that each contained a randomly generated registration code for participation in the online survey (Appendix A). Pseudonymized data were collected using the electronic data capture system secuTrial^®^ and stored on a protected server infrastructure (located in Germany) that was approved for the storage of patient data. The survey consisted of 39 multiple choice questions, with the option to skip individual questions or give alternative answers as free text, and was accessible online from 15 December 2020 to 31 March 2021. The questionnaire was developed by physicians active in treating PD patients; yet, due to the acute development of the pandemic, no pretesting or validation process of the questionnaire was performed [20,21,22]. To preserve the anonymity of the participants in this highly specialized field, no demographic or regional data were assessed. All questions were written in German. The study was approved by the Ethics Commission of the Technical University of Munich (TUM), Germany, under the number 746/20 S-EB, and informed consent was obtained online. The detailed questionnaire is available as Appendix A. This manuscript was prepared according to the Checklist for Reporting of Survey Studies (CROSS) guidelines (Appendix A) [23].

### 2.2. Statistical Analysis

Data analysis was performed with SPSS version 26 (IBM, Armonk, NY, USA) and R Version 4.1.0 (R Core Team, Vienna, Austria). No respondent was excluded, but individually missing data for certain questions were excluded from the analysis. For all closed-ended questions, absolute and relative frequencies are given for all categories. Open-ended or free-text answers were combined into categories (e.g., workplace) and all individual answers are additionally provided as Appendix A. For evaluation of associations between two categorical variables, Fisher’s exact test was performed and odds ratios with 95% confidence intervals were estimated. In addition to these analyses, logistic regression models were fitted to the data to account for potential confounding variables. Due to the small number of events of interest [24], different regression models were fitted subsequently including the variable of interest and one potential confounding variable. Only variables with absolute frequencies of at least ten in the smallest category were considered as potential confounders. Odds ratios with 95% confidence intervals are presented for the covariate of interest.

## 3. Results

Thirty-nine neurologists and one physician with an unspecified specialization participated in this survey. Of the respondents, 75.0% reported that they mainly worked in a hospital, 10.0% in an outpatient department, and 12.5% in a private office; 95.0% claimed that they regularly treated PD patients, and for 72.5%, PD patients represented their main patient group. One participant exclusively treated PD patients. 

Overall, 97.5% of respondents described the medical care for PD patients as worse compared to the time before the pandemic (Figure 1A). The participants’ own subjective quality of treatment deteriorated “to some degree” for 37.5% (Figure 1B); 40.0% found their work with PD patients to be more demanding. A total of 42.5% treated PD patients less frequently, while 7.5% treated them more often. A total of 87.5% had to cancel outpatient visits and 97.5% reported that outpatient visits were cancelled by patients (Figure 1C). Of the respondents, 70.0% closed their offices at least partially and 5.0% closed entirely at some time point during the pandemic. Most participants had difficulties organizing therapy appointments for their patients (speech therapy: 95.0%, physio- or occupational therapy: 92.5%, rehabilitative measures: 94.6%, Figure 1D). Similarly, most participants had difficulties organizing specific treatments, such as endoscopic interventions (e.g., for levodopa carbidopa intestinal gel (LCIG) treatment) (67.5%), deep brain stimulation (DBS) surgery (71.1%), and referrals to a hospital (83.8%). For one participant, it was impossible to organize DBS surgery. To compensate for the deficits in medical care, routine treatment was supplemented with alternative options, primarily with telephone (77.5%) and online (64.1%) consultations (Table 1). Only telephone consultations were “broadly accepted” by patients, as reported by 54.8% of respondents, while all other alternative options were described as “somewhat accepted” by most participants. 

In addition to the impact on routine treatment, 82.5% reported a worsening in the severity of symptoms in their PD patients, 60.0% observed a symptomatic decline in “some” patients, and 22.5% in “many” patients (Figure 1E). Symptom deterioration was observed across all functional domains (neuropsychiatric (82.5%), motor (75.0%), sleep-associated (71.1%), and autonomic (23.7%) symptoms) (Figure 1F). Notably, no participant reported an improvement in symptoms. 

A total of 62.5% of respondents treated COVID-19 patients. Overall, 55.0% felt “at risk” concerning their health due to their profession. A total of 57.5% felt “very” adequately protected by available protective measures, while 42.5% found them to be adequate “to some degree”. No participant found protective measures to be inadequate. 

We found no evidence that physicians who were active in treating COVID-19 patients felt more at risk concerning their health than those who were not (52% “treating” vs. 60% “not treating”, *p* = 0.7470, univariate odds ratio (OR) = 0.72, 95% confidence interval (CI) 0.19–2.62), with a very similar association after adjusting for the need to close their office during the pandemic (OR = 0.78, 95% CI 0.20–2.91, *p* = 0.708) or a change in treatment frequency of PD patients (OR = 0.73, 95% CI 0.19–2.65, *p* = 0.630). Furthermore, physicians who treated COVID-19 patients did not feel less adequately protected (32% “treating” vs. 60% “not treating”, *p* = 0.1074, univariate OR = 0.31, 95% CI 0.08–1.16), even after adjusting for the need to close their office during the pandemic (OR = 0.34, 95% CI 0.08–1.36, *p* = 0.133) or a change in treatment frequency of PD patients (OR = 0.31, 95% CI 0.08–1.15, *p* = 0.087). Treating COVID-19 patients did not lead to an excessive deterioration of the respondents’ quality of care for PD patients (60% “treating” vs. 67% “not treating”, *p* = 0.7458, univariate OR = 1.33, 95% CI 0.36–5.37), which was also the case after adjusting for a change in the treatment frequency of PD patients (OR = 1.48, 95% CI 0.38–6.32, *p* = 0.575), closing their office (OR = 1.31, 95% CI 0.34–5.44, *p* = 0.696), implementation of online tools (OR = 1.58, 95% CI 0.38–7.26, *p* = 0.536), or subjective increase of “stressful” work (OR = 1.28, 95% CI 0.33–5.19, *p* = 0.722). 

In general, hygiene measures required additional work for respondents’ practices (Figure 1G). For 51.3%, the pandemic had a negative financial impact (Figure 1H) and 34.2% had additional expenditures due to the pandemic (Figure 1J). A total of 97.5% wished to be vaccinated against SARS-CoV-2, while all physicians would recommend a vaccination to their PD patients. “Lack of trust” was given as one participant’s reason for their rejection of the vaccine.

## 4. Discussion

In contrast to most previously published analyses on the impact of the COVID-19 pandemic on PD patients [8,17,18,25,26,27,28,29,30,31,32,33], we present here the perspective of their attending physicians. We invited physicians from 67 expert health care facilities, 40 of which participated, resulting in a comparably high response rate [34].

Overall, 75% of the participants reported that they mainly treated PD patients, confirming a high degree of specialization and therefore providing a representative view on the state of PD patient care during this time in Germany. Respondents were largely in agreement that their PD patients’ symptoms worsened during the pandemic, which is consistent with studies of PD patient cohorts [8,18,30]. The reasons for this are multifaceted, but include a reduction in physical activity since pandemic onset, the cancellation of individual or group activities, and a lack of optimization of drug therapies due to cancelled appointments [8,10,35]. A decline in motor symptoms was observed by 75% of the participants. Additionally, more than 80% of the physicians in our study reported a deterioration in neuropsychiatric symptoms, exceeding findings from other groups [32,36]. At the same time, 97.5% reported a deterioration in medical care for PD patients. Most of the physicians had to cancel appointments and treatments, either on their own initiative or at the request of patients. From 22 March to 4 May 2020 (the first lockdown period in Germany), the number of initiated LCIG therapies significantly declined [10]. Two-thirds of the participants in our study confirmed they had difficulties organizing endoscopic interventions (e.g., for LCIG) or DBS surgery for their patients. Although many physicians attempted to compensate for cancelled outpatient visits with telephone or online visits, the acceptance rate of patients, particularly of online visits, was low. This contradicts findings from the U.S., where the use of synchronous video conferencing for the treatment of PD patients significantly increased since pandemic onset and was broadly accepted [34]. The low acceptance rate of digital tools among patients in Germany is in line with the low degree of digitalization in the German healthcare system [18,37], as patients are less acquainted with the use of such tools for online appointments. Therefore, physicians have to remember that the use of digital remote visits may not be suitable for all PD patients [38]. In addition, over 90% of the physicians claimed that they had problems organizing outpatient therapies. This is consistent with data from other groups, which indicated that the majority of PD patients reduced their physical exercise and participation in group activities during the pandemic [8,11]. Despite the high degree of specialization of the respondents, nearly two-thirds of the participating physicians treated patients with COVID-19. Even though 55% of the participants felt “at risk”, most of the physicians felt adequately protected by the available hygiene and safety measures, including those who actively treated COVID-19 patients. Despite a law passed by the federal government of Germany to offset COVID-19-related financial burdens on healthcare facilities [39], over 50% of respondents reported a reduction in income to some degree and one-third had additional expenditures due to the pandemic. In addition, 40% of surveyed movement disorder specialists found their work with PD patients more demanding since the outbreak of the pandemic, regardless of whether they treated COVID-19 patients. This is confirmed by surveys of physical therapists [40] and other healthcare professionals [41] who noted an increase in psychological and work-related stress.

Our study has clear limitations. Most of the participants were contacted via specialized centers, and physicians who see PD patients less often were under-represented. Because surplus resources are lacking for these physicians, we can speculate that the negative impact may be even more pronounced for their patients. This survey was based on semiquantitative assessments and did not require participants to substantiate their statements with objective data, so the data might be biased by individuals’ perception of the pandemic. Furthermore, this questionnaire did not undergo a validation process and therefore might not cover all aspects of the pandemic’s impact on attending physicians.

However, this study highlights current deficiencies in medical care for PD patients and argues for urgent improvements in digital infrastructure in the German healthcare system to improve patient care. One way to reduce hospitalization rates in the face of necessary interruption of outpatient rehabilitation services due to pandemic-related contact restrictions is to provide telerehabilitation support [42] that ensures continuous monitoring of patients and improves not only the health status but also the quality of life of PD patients.

## Figures and Tables

**Figure 1 brainsci-12-00353-f001:**
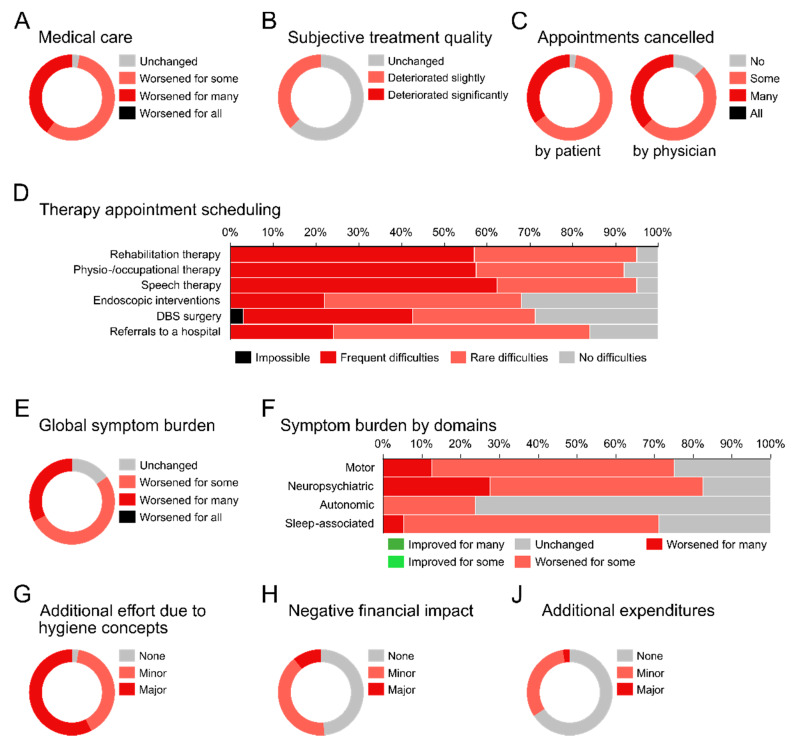
(**A**) Change in medical care quality compared to prepandemic period. (**B**) Change in subjective treatment quality compared to prepandemic period. (**C**) Cancellation of appointments by physician or patient. (**D**) Difficulties physicians faced in scheduling therapies for PD patients. DBS, deep brain stimulation. (**E**) Change in the overall symptom burden. (**F**) Change in motor and non-motor symptoms observed by study participants. (**G**) The implementation of hygiene concepts required additional effort from physicians. (**H**) Financial impact physicians faced since pandemic onset. (**J**) Additional expenditures for physicians during the pandemic.

**Table 1 brainsci-12-00353-t001:** Alternative medical care options for PD patients during the pandemic. Participating physicians offered several alternative medical care options, which were received differently by patients.

Result			No (%)	
Home visits to patients				
	No		34 (87.2)	
	Yes		5 (12.8)	
		Not accepted		0 (0)
		Somewhat accepted		3 (60.0)
		Broadly accepted		2 (40.0)
Online consultations				
	No		14 (35.9)	
	Yes		25 (64.1)	
		Not accepted		1 (4.0)
		Somewhat accepted		20 (80.0)
		Broadly accepted		4 (16.0)
Telephone consultations				
	No		9 (22.5)	
	Yes		31 (77.5)	
		Not accepted		0 (0)
		Somewhat accepted		14 (45.2)
		Broadly accepted		17 (54.8)
Written information				
	No		33 (84.6)	
	Yes		6 (15.4)	
		Not accepted		0 (0)
		Somewhat accepted		5 (83.3)
		Broadly accepted		1 (16.7)
Other				
	No		37 (92.5)	
	“Homepage and social media”		1 (2.5)	
		Broadly accepted		1 (100)
	“Emergency consultation hour”		1 (2.5)	
		Broadly accepted		1 (100)

## Data Availability

The data presented in this study are available in the manuscript and Appendix A.

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
