# Peer review of "Impact of the COVID-19 Pandemic on Patients with Parkinson’s Disease from the Perspective of Treating Physicians—A Nationwide Cross-Sectional Study"

_brainsci, 2022, doi:10.3390/brainsci12030353_

Round 1

Reviewer 1 Report

Thanks for the opportunity to review. The manuscript is captivating and concise, but in this regard it has some methodological gaps that need to be addressed. I recommend following guidelines to complete it, hypothesizing a hierarchical regression study based on age or evaluating the regional impact. Finally I recommend if you can include the questionnaire in the manuscript

Abstract: please enrich the abstract with methodology, how many subjects were reached? Was a regression study conducted? What physical results and therefore a conclusion

Introduction: Seems a bit bare, enrich the background with an overview of the need for continued and appropriate rehabilitation cycles for these patients

regular “outpatient” visits

Here, we present data from a nationwide online survey of specialists involved in the treatment… “In this scenario, this cross-sectional survey aimed to ..

Methods: please follow CROSS checklist (https://www.researchgate.net/publication/353522424_Checklist_for_Reporting_Of_Survey_Studies_CROSS)

Participants section totally missing

Discussion

In contrast to most previously published analyses on the impact of the COVID-19 pandemic on PD patients.. references missing

Reviewer 2 Report

  1. The reviewer would like to know why this manuscript was uploaded as communication? Is it a preliminary study?
  2. The abstract is missing information about the aim/objectives and methodology.
  3. It is advised to separate the paragraphs in the introduction. 1st about Germany and global information. 2nd about movement disorders and possible influence by the COVID pandemics. 3rd specific about PD and COVID. 4th about the aim of the study.
  4. The questionnaire performed was validated or the authors can provide validation for its use?
  5. Statistical analysis. It is advised assistance for the description of this section.
  1. How were assessed open and close-ended questions?
  2. How was data distributed?
  3. Describe the power of the study.
  4. Could the authors provide the spreadsheet of the individuals?
  5. Was all categorical data evaluated by Fischer’s?
  6. Could the authors perform a multivariate analysis? Or perform a better description of possible confounding variables
  1. Figure 1. Data should be described in tables. If a figure is needed, it is advised to provide column graphs bar only.
  2. The results and conclusion sections were described as massive one-block. It is advised division in paragraphs or opinion of the editor for this design.

Round 2

Reviewer 1 Report

The manuscript has improved greatly, now it has a more solid structure .. Change statistics with “Statistical Analysis”. I would increase the quality of the figure which can be very captivating for readers and other authors. I would also add some significance data, and if anything, I would duplicate it in a more accessible/user-friendly format for an graphical abstract… These are just guesses. I suggest only a minor revision of the paper.

I could suggest a different end of discussion:
“However, this study might highlight current gaps in medical care for people with PD, and our findings suggest urgent improvements in the digital infrastructure of the German healthcare system to enhance the care for these people. With the necessary interruption of outpatient rehabilitation services due to the first COVID-19 pandemic lockdown, it might be helpful to reduce hospitalization rates by providing telerehabilitation assistance, guaranteeing continuous patients' monitoring, to improve not just the state of health but their quality of life [1]. Taken together, the results of the study focused on the importance of telehealth that could remove the pandemic burden from clinicians [2] without affecting the patients, but which thus focuses not only on cure but on patient care.”

(ref: 1= http://doi.org/10.1108/jet-11-2020-0047 , 2= https://doi.org/10.3390/ijerph18189676 )

Reviewer 2 Report

1) Information about the validation of the study should be disclosed in the ‘‘limitations’’ section. Also, a similar statement from the literature should be referenced to provide scientific evidence.

2) The reviewer believes that the ‘‘statistical’’ section still needs improvement. It is advised professional assistance for a better description of how did the authors analyze the data?

3) In the discussion, could the authors provide a table comparing before and after COVID-19 main features? This would highly impact the quality of the manuscript.
